# The Potential of Gamification for Social Sustainability: Meaning and Purposes in Agri-Food Industry

**Maria Elena Latino** [1,*] **, Marta Menegoli** [1] **, Fulvio Signore** [2] **and Maria Chiara De Lorenzi** [1]

1   Department of Innovation Engineering, University of Salento, 73100 Lecce, Italy;
    marta.menegoli@unisalento.it (M.M.); mariachiara.delorenzi@unisalento.it (M.C.D.L.)
2   Department of Humanities, Letters, Cultural Heritage and Educational Studies, University of Foggia,
    71121 Foggia, Italy; fulvio.signore@unifg.it
*   Correspondence: mariaelena.latino@unisalento.it; Tel.: +39-0832-297949

**Abstract:** Nowadays, digital platforms are applying some game-linked elements in their services with the aim to attract, retain and shape human interaction. Therefore, it is essential to investigate gamification with regard to its social sustainability. Gamification strategies are strategical in the agri-food industry to reach social and marketing goals. Despite the presence of several literature reviews on gamification, no study proposes a reflection on the meaning and purposes of gamification in the agri-food industry. This paper aims to identify the core dimensions underlying the concept of gamification, analysing its definitions and purposes through a systematic literature review, content analysis and principal component analysis. Eight core dimensions were detected leading to the conception of a new definition of gamification in the agri-food industry. Our results highlighted the potential of gamification to impact society, leaving points of reflection on how it can be made more inclusive and effective. Theoretical and practical implications were provided for academics, researchers, agri-food companies, policy makers, management engineers, technology makers, marketers and practitioners. The novelty of this study lies in the application of a social psychology methodology to give meaning to the words, overcoming the limits derived from qualitative research based on, only, content analysis.

**Keywords:** gaming; game; meaning extraction method; qualitative analysis; systematic literature review; learning

## 1. Introduction

Nowadays, most digital platforms apply some game-based logics intended to attract, retain and shape human interaction. It hence becomes essential to investigate gamification with regard to its social sustainability, "a concept describing conditions fostering the well-being and development of communities" [1], p. 1. Gamification is usually linked to the concept of games which are forms of play capable of creating and enhancing enjoyment and a sense of achievement [2]. However, gamification is able to serve also non-game purposes, providing a digital business environment based on rules and motivators [3–5] for engaging consumers through interactive experiences and affecting their behaviours [6,7]. In the industrial context application, gamification is defined as the use of game design elements and the development of mechanisms to involve people [8], motivate actions, promote learning [9,10] and solve problems [11,12]. The gaming industry has, currently, seen the most extensive developments in medicine, education, travel, entertainment, heritage and especially the marketing sector [13,14]. Indeed, by leveraging marketing strategies, businesses can promote the purchase of products, engaging more consumers and ensuring more market share through gamification processes and applications [15]. According to the marketing viewpoint, gamification is helpful in changing attitudes and behaviours fostering value creation for companies [16,17]. Indeed, it is able to use stimuli and gamified elements (such as rewards and badges) that are properly designed with the help of innovative technologies (e.g., virtual and augmented reality) to motivate consumers to follow rules

and perform planned behaviours [18,19]. Therefore, the main achievable benefits are the engagement, the motivation and the involvement of users, triggering the (i) learning processes that users experience as a game, albeit not replacing education itself, but rather improving it [20–22], and (ii) planned behavioural processes [23–25]. However, the authors of [26] discussed that not all gamification efforts yield the desired results and that boundary conditions (i.e., meaningfulness and disclosure) could help managers avoid the potentially detrimental effects of gamification.

Overtime, several studies summarising the gamification research field have proposed literature reviews with different purposes and industries (e.g., agri-food, medical). The authors of ref. [27] conducted a systematic review to assess the effectiveness, through an empirical lens, of gamification in the health and wellbeing domain, discovering that gamification can have a positive impact targeting behavioural outcomes, particularly physical activity. The review in [28] revealed that gamification could represent a crucial way, in the medical field, to provide opportunities for risk-free clinical decision making, distance training, learning analytics and, swift feedback. The authors of ref. [29], focusing on the agri-food and social science fields, collected the existing literature examining household consumption and pro-environmental behaviours to classify behavioural interventions targeting the household food–energy–water nexus, basing their research on gamification also. The authors or ref. [30] performed a systematic meta-review of the theoretical foundation of gamification, identifying 118 theories in the areas of motivation, behaviour, and learning used to explain gamification. The authors of ref. [31] presented a review to identify game design mechanics and features that are reported to commonly influence consumer behaviour change, during and/or after gamification interventions in a general application context. The systematic literature review proposed by the authors of [32] explored the distinctions between three evaluation models leveraging gamification, describing the game design and development processes as necessary to create a functional game-based assessment for employee selection. The review in [33] was an analysis of the relationship between gamification, motivation, and learning, and thus, provides pedagogical and didactic ideas for its implementation in non-university education. The authors of ref. [34], thanks to a systematic literature review, analysed gamification for the service field, identifying 34 empirical articles that reflect gamification conceptualisation and can be connected to relevant service research themes (e.g., customer participation, experience, and loyalty).

As emerges from this analysis, the extant literature reviews on gamification are focused on systematising the theoretical foundation of gamification, consumer behaviours during the implementation of gamification strategies, impacts of gamification on learning, benefits from gamification adoption, the game design mechanism and features, the conceptualisation of gamification in the service field and related functions. To the best knowledge of the authors, no study proposes a reflection on the meaning of gamification in order to discuss the key concepts that revolve around this theme. However, synthesising the existing knowledge base, according to this perspective, represents an opportunity to create a shared overview of gamification, useful for guiding and supporting marketing academics, practitioners and managers in the definition of related marketing strategies. In order to fulfil this emerging gap, this paper aims to identify the core dimensions that underlie the concept of gamification. In doing this, the meaning of gamification, its purposes and the transversal core dimensions that the gamification concept assumes were investigated. In order to increase the validity of the results and implications of this study we choose to focus the study on a specific industry: agri-food. This choice was guided by the relevance of the agri-food products and services to the life of average consumers and the impact of their production on environmental and social sustainability which leads companies and governments to assume an active and responsible role in directing aware consumer behaviour, also leveraging gamification strategies (more details in Section 2).

In this study, gamification definitions and purposes were discovered through a content analysis of a body of knowledge discovered through the systematic literature review method. Then gamification definitions were analysed according to the meaning extrac-

tion method (MEM), based on principal component analysis (PCA). Specifically, the last method allows the processing of a text by interpreting it through the certain latent variables outlined by it. Words, therefore, become the main vehicle through which to identify underlying meanings that can explain a given variance value, with the aim of reducing residual variance as much as possible. In this study, we use the extracted words from the gamification definitions identified in the sample to quantitatively identify transversal core dimensions of meaning that allow the categorisation of such word texts. Eight core dimensions were discovered: "the social-motivational change introduced by gamification", "the health educational role of gamification", "the co-creation processes of gamification", "the reward mechanism of gamification", "the monitoring mechanism in gamification", "real-life contextual procedure", "the stakeholder's supply chain engagement", and "the functional gaming purpose". Some of these are new compared to the contents that emerged from the first phase of the analysis (the content analysis), enriching the understanding of the research field and confirming the presence of an undeclared meaning in the definitions. These represent the advancement of knowledge that this study brings to the research field. Indeed, the novelty of this study is the application of a methodology mainly used in social psychology to give meaning to the words used by placing them in general categories [35]. This technique was used to discover underlying meanings of gamifications that were useful to better understand the gamification-based strategies overcoming the limits of the qualitative research based on content analysis. Moreover, in order to better capitalize on the new knowledge emerging from this study, we propose a new definition of gamification in the agri-food industry capable of encompassing and eliciting the discovered core dimensions.

Marketing academics, practitioners and agri-food companies could benefit from the results of this study by increasing the general awareness about gamification and understanding how it could be applied to implement direct marketing strategies, with the outcome of promoting food products and services and related conscious behaviours toward consumers, or indirect marketing strategies that, through the involvement of stakeholders in virtuous supply chain paths or the achievement of better behavioural and operational standards, are capable of positively acting on consumers' trust.

## 2. Materials and Methods

### 2.1. The Protocol for Review

The systematic literature method allows the establishment of knowledge boundaries of a specific research field [36,37], identifying and critically analysing the related studies [38]. The review protocol was established according to the PRISMA flowchart [39] (see also Supplementary Materials) and the 6W framework [40] (Figure 1). The definition of a search scheme composed of keywords [41] and Boolean operators [42] is important to identify the samples that need to be analysed. In this study, the research scheme was established by leveraging the authors knowledge (who) with the aim of identifying studies focused on the application of gamification in the agri-food industry ("gamification" and "food") (how). The choice of this scheme was guided by the authors' desire for a broad selection process, avoiding the loss of important works. The search scheme was used for questioning in the Scopus data base (http://www.scopus.com (accessed on 15 February 2023)) (where). Managed by Elsevier publishing, this is considered an extensive and high-quality data base [43,44]. Queries were entered in the data base in the February 2022 (when) in the "Article title, Abstract, Keywords" field of the Scopus setting without imposing any temporal restrictions. The initial sample was composed of 124 papers. A selection was made according to the following criteria (why): (i) limit to studies in English and, (ii) exclude studies belonging to medicine and nursing research areas. This procedure led to a final sample of 56 papers to be analysed (what).

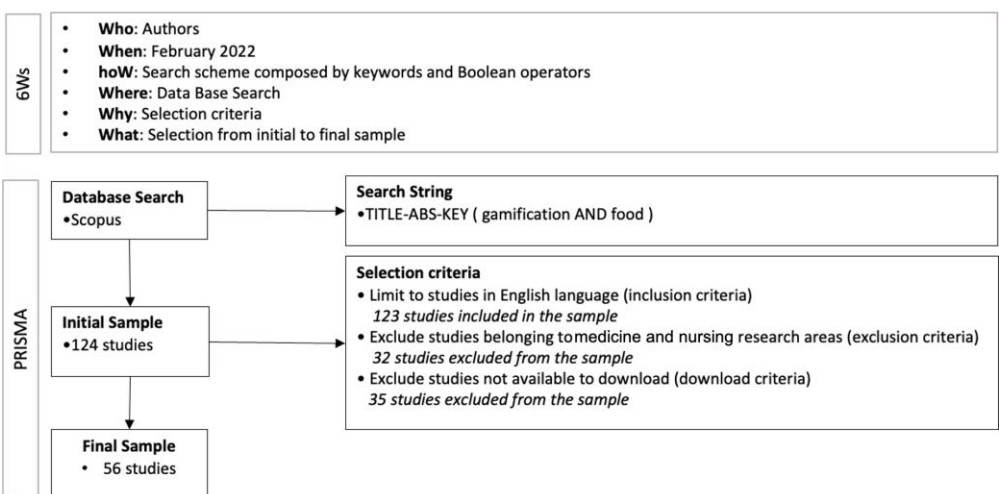

**Figure 1.** The protocol for review.

### 2.2. Qualitative Content Analysis

The sample was firstly analysed using qualitative content analysis to detect the definitions of gamification and the application purposes for the agri-food industry. Qualitative content analysis is capable of systematically classifying and organising contents into categories that describe the topics under analysis [45]. From this analysis, 34 definitions and 5 purposes of gamification were recognised. These definitions were analysed, identifying (i) the authors that provided the definition, (ii) how gamification is considered (e.g., the strategy, method, and approach), hereinafter referred to as identification and, and (iii) the context in which gamification is applied (e.g., education, food choices, and lifestyle). The research team acted as coders of the qualitative content analysis after the implementation of a training phase that was needed to avoid differences in coding and abstraction, to minimize human errors and to increase the replicability of the procedure [46,47]. Each author analysed the studies in the sample and categorised the related contents using a common Excel scheme. After that, a comparison among the authors' results was made. In the event of conflicting results, a discussion was initiated involving two gamification experts belonging to the authors' institution; adopting focus group techniques, the studies in question were analysed again, and agreement was reached on the categorisation of the contents.

### 2.3. The MEM: PCA on Text

The MEM is a dimension reduction approach applied in the study of complex phenomena with the aim of creating a certain number of interpretable themes starting from text data. The process is based on two steps: a pre-processing operation and a multivariate analysis through PCA. The first phase can be automatised through several software, such as meaning extraction helper (hereafter MEH), which convert single words or phrases into single units. The aim of PCA is to minimize the information lost (e.g., residual variability), maximising the retained information (e.g., variability explained). It is a multivariate analysis technique that aims to reduce the complexity of the analysis of a phenomenon by creating new latent variables, known as principal components, that can explain the case while preserving the highest quantity of information about it. Specifically, in this study, MEH is applied to analyse the gamification definitions with the aim of identifying the transversal dimensions that underlie this concept. The transversal dimensions, or components, are latent structures that can be considered to have a general meaning underlying the terms they are composed of. For this reason, after carefully reinterpreting the emerging components and the words affecting them, a qualitative exploration of them was conducted in order to identify cross-dimensional meanings that could capture the meaning of what emerged through statistical analysis [48]. This bottom-up technique is widely applied in

social psychology [48,49] and all those areas in which the aim is to analyse content from large-scale language data.

Since MEH allows the conversion of phrases into single units, using the Jamovi software (version 1.8.1), the matrices useful for the analysis (e.g., frequency and binary matrixes) were extracted; the presence of words was identified with the value 1 and their absence was identified with the value 0 (Figure 2). For a cleaner and more coherent analysis, stop words were deleted from the final dataset of definitions.

| gamification | defined | using | game | design | elements |
|---|---|---|---|---|---|
| 1 | 1 | 1 | 1 | 1 | 1 |
| 0 | 0 | 0 | 0 | 0 | 0 |
| 1 | 0 | 0 | 0 | 0 | 0 |
| 0 | 0 | 0 | 0 | 0 | 0 |
| 1 | 0 | 0 | 0 | 0 | 1 |
| 0 | 0 | 0 | 0 | 0 | 0 |
| 1 | 0 | 0 | 0 | 0 | 0 |
| 0 | 0 | 0 | 1 | 1 | 1 |
| 1 | 0 | 0 | 1 | 0 | 0 |
| 1 | 0 | 0 | 0 | 1 | 1 |
| 0 | 0 | 0 | 0 | 0 | 0 |
| 0 | 0 | 1 | 0 | 1 | 1 |
| 1 | 0 | 0 | 0 | 0 | 0 |
| 1 | 0 | 1 | 1 | 0 | 0 |
| 1 | 0 | 0 | 1 | 1 | 1 |
| 1 | 0 | 0 | 0 | 0 | 0 |
| 1 | 0 | 0 | 0 | 0 | 1 |
| 1 | 0 | 0 | 1 | 0 | 1 |
| 1 | 0 | 0 | 0 | 0 | 0 |
| 0 | 0 | 0 | 0 | 0 | 0 |

**Figure 2.** Draft for the binary matrix adopted for the subsequent analysis.

In order to identify the core dimensions that underlie the gamification strategies in the agri-food industry, PCA was then performed upon a corpus of 253 words extracted from the 34 definitions. KMO = 0.55 and Bartlett's test < 0.001 confirm the suitability of the starting dataset.

Based on the analysis output, the number of chosen dimensions was 8. Although the extracted dimensions with eigenvalues greater than 1 were greater than 8, the number of final dimensions identified was 8 because the meanings of the subsequent dimensions overlapped. Therefore, one of the limitations of textual analysis is the generative dimension of words and the fact that synonyms very often appear. The selected requirements were not determined by evaluating the standard criteria for the identification of the components in the PCA (e.g., an eigenvalue greater than 1, an explained variance of around 70% and maximum variance discarded in the scree plot), but with the intention of simplifying the phenomenon and, at the same time, keeping unchanged the meaning of the dimensions. The rotation algorithm identified was varimax, as the aim was to use dimensions that were not related to each other, i.e., that could explain different amounts of information and variance. The 8 dimensions chosen explained a total of 33.7% of the sample, and although this percentage could be considered small for most factor analyses of questionnaires, it is not the case for natural language use, as suggested by [50]. To facilitate the reading of the results and to appropriately differentiate the latent factors that emerged, we proceeded by considering a loading of ≥0.60 as a limit (Table 1).

**Table 1.** Brief excerpt of the words influencing some of the dimensions most to detect transversal meanings of gamification strategies in the agri-food industry, based on the PCA technique.

| Words | Components | | | | | | | |
|---|---|---|---|---|---|---|---|---|
| | 1 | 2 | 3 | 4 | 5 | 6 | 7 | 8 |
| potential | 0.98 | | | | | | | |
| changes | 0.98 | | | | | | | |
| friendly | 0.98 | | | | | | | |
| customer | 0.98 | | | | | | | |
| loyalty | 0.98 | | | | | | | |
| belief | 0.98 | | | | | | | |
| collaboration | 0.98 | | | | | | | |
| employee | 0.98 | | | | | | | |
| competition | 0.98 | | | | | | | |
| trend | 0.75 | | | | | | | |
| performance | 0.73 | | | | | | | |
| motivation | 0.71 | | | | | | | |
| encourage | 0.73 | | | | | | | |
| participation | 0.73 | | | | | | | |
| eat | | 0.97 | | | | | | |
| vegetables | | 0.97 | | | | | | |
| children | | 0.97 | | | | | | |
| willingness | | 0.97 | | | | | | |
| berries | | 0.97 | | | | | | |
| integrated | | 0.97 | | | | | | |
| food | | 0.97 | | | | | | |
| taste | | 0.97 | | | | | | |
| activities | | 0.80 | | | | | | |
| education | | 0.80 | | | | | | |
| increase | | 0.60 | | | | | | |
| new | | | 0.96 | | | | | |
| producers | | | 0.96 | | | | | |
| create | | | 0.96 | | | | | |
| snack | | | 0.96 | | | | | |
| co-creation | | | 0.96 | | | | | |
| developed | | | 0.96 | | | | | |
| consumers | | | 0.73 | | | | | |
| products | | | 0.72 | | | | | |
| tool | | | 0.72 | | | | | |
| healthy | | | 0.67 | | | | | |
| achievements | | | | 0.96 | | | | |
| badge | | | | 0.96 | | | | |
| competitiveness | | | | 0.96 | | | | |
| goal-oriented | | | | 0.96 | | | | |
| actions | | | | 0.96 | | | | |
| incorporate | | | | 0.96 | | | | |
| rankings | | | | 0.96 | | | | |

These criteria of loading enabled us to clearly distinguish the influencing terms, as they constitute an easily intelligible and openly interpretable matrix. The final elaboration provided 71 words (28.1% of the initial corpus) classified, through linear combination, into 8 dimensions of defined significance.

### 3. Results

A first level of knowledge emerges from the qualitative analysis of the sample; 34 definitions of gamification were discovered during the period 2009–2021 (see Appendix A). An average value of about two definitions per year are discovered. The first definition of gamification arose in 2009 from [51]. The peak of published gamification definition occurred in 2011 when several authors enriched the research field proposing four new defi-

nitions [11,12,52,53]. After that, the trend remained almost constant, with the contribution proposed in [27,54–61]. Only one new definition appeared in the 2021 [62] and, since the analysis sample was retrieved in February 2022, no new definitions of gamification were published in the first months of this year.

The most recurrent definition is the one proposed in [11,12] leading to the most common identification given to gamification: the use of game design elements particularly in non-game contexts. Other identifications of gamification include that it is a tool for social engagement, strategy, approach, set of activities, or services [63]. With reference to the agri-food industry, the definitions of gamification were adopted in different food-related contexts. The main one was the possibility to enact, through gamification, a behavioural change in food choices and consumption, but also within other contexts which involved education, healthcare, learning, engagement, and finally technological changes. Although gamification has been studied and applied with several meanings and in different contexts, it aims to meet common purposes or needs in the agri-food industry. Firstly, it raises community awareness about healthy nutrition in order to preserve health and wellbeing. Moreover, gamification is aimed at encouraging sustainability in all its three dimensions (economic, environmental and social) following educational purposes [64]. At the same time, gamification could be applied to encourage the use of technologies in the social and industrial production fields, to control and monitor, even remotely, the real-time production cycle. The use of gamification for brand engagement purposes and consumer profiling in the marketing area is less commonly debated in the current literature. Finally, gamification has been exploited in a learning context at the workplace for employee training and monitoring purposes.

Based on the above-mentioned methodology and on the analysis text corpus, a brief descriptive outline of the results is shown in Figure 3. By considering the limitations of the word elaboration approach suggested by [50], words extracted from the literature research explained a good-enough variance (34%).

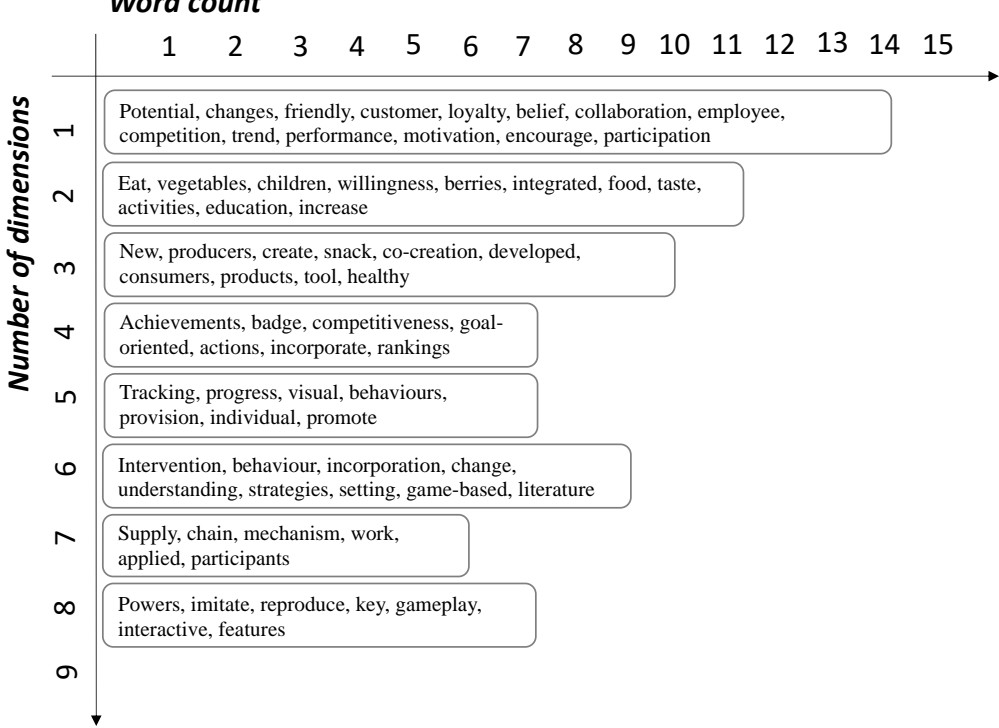

**Figure 3.** Word count and words for each dimension.

Analysing these results according to a descriptive perspective, the research team recognised in the first dimension, predominantly influenced by terms such as motivation,

encouragement, participation, collaboration, loyalty, changes, and customer, a strongly applicative tendency focused on the aim of inducing a change, including a behavioural one, based on a participative social motivation. Therefore, we propose to define this dimension as "the social-motivational change introduced by gamification" with the aim of underling the role that this strategy plays in encouraging collaborative and participating behaviours between customers and agri-food operators. The second dimension, on the other hand, is characterised by terms such as food, education, berries, children, and vegetables; here, therefore, the focus is on the specific recipients of the gamification strategies (children and education) and the specific elements to be conveyed (vegetables, food, and education activities). For this reason, the dimension was defined as "the health educational role of gamification" with the aim of underling the capability of this strategy to increase the awareness of healthy eating behaviours, especially in children. The third dimension is based on terms such as increase, create, co-creation, consumers, products, healthy, and tools. It underlines that gamification strategies can involve consumers in the process of product co-creation, allowing the company the opportunity to bring to the market more desirable products. It refers also to how the food information to be transmitted to the consumers, that may be formed according to inter-disciplinary mechanisms (e.g., psychological [65] and marketing), leverage on co-construction processes focused on increasing the awareness of food quality and health topics. Therefore, we propose to call this dimension "the co-creation processes of gamification". The fourth dimension is influenced by terms such as goal-oriented, rankings, badges, competitiveness, and achievements underling the tools generally applied during the gamification strategies to involve users. Therefore, the presence of the terms goal-oriented and competitiveness referred to the players and identified the nature of this game-based strategy. During the establishment of gaming strategies, companies could use ranking, badges, and achievements as reward mechanisms. Thus, this emerging dimension was called "the reward mechanism of gamification". The fifth dimension is contingent on terms such as tracking, progress, and provision. It seems to lead to a factor marked by the intention to monitor the progress that is achieved through play-based, agri-food-oriented educational methods. The name of this dimension was, therefore, "the monitoring mechanism in gamification". The sixth dimension identifies terms such as intervention, behaviour, incorporation, change, understanding, strategies, setting, game-based, and literature and seems to refer to a real-life intervention perspective, based on the human understanding and the use of game-based strategies and is geared towards improvement through the analysis of the relevant scientific literature. Hence, the proposed name of the latent dimension was "real-life contextual procedure". The seventh dimension focuses on terms such as supply chain, mechanism, work, applied, and participants, and envisages, as in the first explicit dimension, the proposition of strategies that presuppose involvement through active mechanisms with the actors forming the supply chain. As a result, the name of this dimension is "the stakeholder's supply chain engagement". Finally, the last dimension elicits a component with a distinct emphasis on a functional dimension of gaming, a psychological strategy with a strong applicative inclination relying on mechanisms such as imitation, reproduction, and interaction (terms such as powers, imitate, reproduce, interactive, and gameplay). Thus, this last dimension was defined as "the functional gaming purpose".

## 4. Discussion

This study provides a clearer framework of gamification definitions and purposes and the transversal dimensions that underlie the concept of gamification in agri-food industry applications. Referring to the features that characterize the existing definitions of gamification, we found a wide diffusion of the definition proposed in Deterding, S.'s works [11,12]. This evidence leads the way in considering the relevance of some authors, regarding the theme, in the scientific panorama as pillars in specific matters and their works as a starting point for future research routes. In witness whereof, several authors referred to Deterding S.'s gamification definition [11,12] to identify the concept of design

elements to be employed in user engagement strategies [66–79] and the application context of the non-gaming context [29,59,69,73,80–82]. However, this represents a weakness of the research field, due the presence of other definitions capable of providing interesting types of identification of the gamification concept. For example, gamification is understood as a strategy or an approach [59,83,84], an emerging trend [57] or a set of activities or services [67,84,85] that are useful for providing learning, communication and interaction to the user in a fun and engaging atmosphere in order to enhance results and behaviours, providing a better user experience. Moreover, gamification is understood as an integration or supporting tool (i) to foster co-creation processes of, for example, a new kind of healthy food product as well as a form of product advertising [86,87]; (ii) to engage specific users such as consumers, children and farmers [56,88–92]; (iii) to stimulate food behaviour change [29] and technology usage [55]. Therefore, the purpose for which gamification is applied in the existent studies emerged clearly. Based on our results, we can confirm for the agri-food industry, the same trends recognized by the authors of [21,25] of the purposes of gamification applications in the general industrial context. In particular, following the suggestions of [21,25], gamification falls into the social context considering the motivation and the involvement of users in its main purposes and, triggering (i) learning processes that users experience as a game, albeit not replacing education itself, but rather improving it [93] and (ii) planned behavioural processes. Gamification in the agri-food industry mirrors this trend, tailoring each of these processes to the agri-food, health and nutrition, wellbeing fields. Gamification is capable of triggering food behavioural change promoting health and nutrition habits in line with more healthy food choices, confirming the results obtained by the authors of [94]. It can also trigger behavioural change with regard to brand awareness, pushing users toward specific products in market, and with regard to technological usage habits for encouraging users who are not digitally native toward an evolution of their own technology acceptance and stimulating digitally native users to maintain or increase their habits regarding technology usage; gamification is capable of addressing educational principles toward more sustainable practices in learning and working contexts. However, that is not all; our analysis, through the MEH method, revealed the presence of eight core dimensions of gamification which the several definitions have in common even if they are not explicitly discussed in them.

Accordingly, the research performed enabled us to derive a second level of analysis of gamification definitions by categorising terms taken from the definitions found. Some of these dimensions confirm the results coming from the content analysis of this study (e.g., the health educational role of gamification), while others provide new interesting, implied findings. Specifically, gamification results as an approach that is useful in the context of collaborative innovation and particularly in co-creation, such as in the interaction and interchange of ideas between users, customers, suppliers and other actors in the development of new solutions [95]. In this way, gamification could be included among the tools of social innovation, which, as proposed by [96], are characterised by certain features, such as the challenges (which are attempted to be solved through the interconnection of various social actors), user participation (through the interactive mode or co-creation mode that modulates social inclusion), the implementation of creative solutions, the processes of empowerment and agency that arise from co-participation, and the aim of changing systems.

The lack of consideration of these aspects and their clear formulation and discussion therefore represent a gap in the current research field that this study fills.

Therefore, the definition of gamification strategies in the agri-food industry involves, in an organised manner, the co-presence of processes, mechanisms, objectives and social actors. As far as processes are concerned, the study showed how in implementing appropriate gamification strategies there is a need to employ co-creation processes, supporting the position of the authors of [97], that are often inter-disciplinary and multi-actor, to improve the information conveyed and the awareness around products and issues. From the viewpoint of gamification mechanisms, our analyses revealed the necessary presence

of reward mechanisms and the monitoring of consumption behaviour. Concerning the objectives, the study made it possible to highlight a propensity of these strategies towards social and motivational change hinging on the educational role often associated with health issues. Finally, regarding the social actors involved, the outputs of the analysis clearly show a tendency towards the application aspect, aimed at real contexts, based on the literature, and the involvement of various stakeholders related to the entire agri-food supply chain. In order to advance the investigated research field, we proposed, according to the discovered core dimensions, a gamification definition for the agri-food industry:

> "Gamification is an action, aimed at implementing a social-motivational change through a co-creation process, with a health, wellbeing and educational perspectives. Rooted on reward and monitoring mechanisms, its applicability is outlined in the real-life and contextual basis, by engaging all stakeholders of the agri-food supply chain with a functional gaming purpose".

## 5. Theoretical and Practical Implications

Leveraging the conception of the new definition and the discovery of the application purposes of gamification, several theoretical and practical implications emerged from this study.

To the best knowledge of the authors, this study is the first to provide a reference framework that summarizes the academic advancements in gamification in the agri-food industry, extending the state of the art. It covers a timespan of about 8 years (2013–2021), and also clarifies the temporal viewpoint of the previous contributions. The eight discovered core dimensions, underlying the meaning of gamification, represent the advancement of knowledge that this study brings to the research field. Indeed, the novelty of this study is the application of a methodology mainly used in social psychology to better investigate the meaning that the gamification strategy has with regard to the agri-food industry, overcoming the limits derived from qualitative research based on content analysis. Moreover, in order to better capitalize on these findings, we propose a new definition of gamification in the agri-food industry that encompasses and elicits the discovered core dimensions.

This result allows us to envisage the concept of gamification in the agri-food industry from an overall perspective and based on elements not previously considered, such as the processes, mechanisms involved, objectives included, and actors mobilised. Using a more extensive approach in gamification analysis may lead to the proposal of a multi-disciplinary context, broadening the perspective and the framework of interpretation of these strategies. This reframing process is based on an extension of the gamification concept, to draw the main attention to the complexity of the logics underlying gamification strategies. For instance, including the involvement of consumers and stakeholders in the definition of gamification strategies could represent a way to develop, through the use of these strategies, products/services that better meet the needs of all those involved in the processes. All these theoretical issues could represent interesting starting points for future research in the gamification field; the advancements underlined in this study for the agri-food industry could be investigated and/or applied in other fields, while the emerging lack in this specific sector can be filled by leveraging the best practices from other sectors (e.g., the creative industries and fashion).

This study also has several practical implications for agri-food companies, agri-food policy makers, management engineers, technology makers, marketers and practitioners interested in gamification applications in the agri-food industry. Companies operating in the agri-food industry could benefit from the results of this study increasing the general awareness around gamification and understanding how it could be applied to implement direct and indirect marketing strategies with the outcome of promoting a food product/service to consumers. Therefore, using gamification to involve stakeholders in a virtuous supply chain or to encourage employees to achieve better behavioural and operational standards represents indirect marketing strategies through which a company (or marketer) can increase their consumer trust level. Moreover, gamification can support

agri-food companies, andmore generally the agri-food industry, in being compliant with the European Union's2030 agenda of achieving Sustainable Development Goals (SDG). Especially, from the social sustainability viewpoint, according to SDG 3, agri-food companies are called on to ensure healthy lives and promote well-being for all people at all ages; gamification can provide support in ensuring health and well-being, enhancing every human's awareness about safe and wholesome food, contributing to the overall wellness of society [29,68,72,88,94,98,99]. According to SDG 4, agri-food companies are called on to ensure inclusive and equitable quality education and promote lifelong learning opportunities for all; gamification can provide support in ensuring quality education on the role of food in health and wellbeing [78,84]. According to SDG 11, agri-food companies are called on to make cities and human settlements inclusive, safe, resilient and sustainable; gamification can provide support in making communities (such as those of students, families with young children, or more generally consumers) more sensible in terms of sustainable behaviours and can stimulate resilient actions, such as those related to food waste management [56,67,77,82,100,101]. On the other hand, food policy makers could find in this study the inspiration to adopt gamification practices with the aim of implementing social campaigns focused on fostering health and well-belling behaviour among citizens and children or fair behaviours for the distribution of value along the players in the agri-food supply chain. Management engineers and practitioners could find in this study a point of reflection on the possibility to implement a wide multi-purpose strategy that leverages gamification, reward and, monitoring mechanisms that could reinforce the competitive position of a company, increasing the level of trust even towards the most suspicious market segments. This study confirms the strategical role of technology in gamification. This could encourage technology makers to design and develop new technological tools, based on innovative and interactive technologies (e.g., augmented reality and the metaverse) or devices (e.g., google glass) to encourage the development and the adoption of gamification. Specifically, the development of low-cost technologies could facilitate the spread of these techniques even among small and micro-enterprises in the sector, which as is known represents a considerable part of the offer. Moreover, since the effects of gamification on consumers have not yet been fully investigated [26], useful hints providing transversal dimensions to build adequate marketing campaigns could serve the industry itself in proposing strategies (that are more sustainable and eco-friendly, for example in [68,102]) that are more targeted and contextualised to the phenomenon. Lastly, valuing the horizontal scalability of our results, other industries worldwide such as those of pharmaceuticals and nutraceutics, which have products similar to food products in terms of interaction with human body (all are introduced into the oral cavity, chewed and/or swallowed, digested, hosted by intestinal transit, absorbed and metabolized), can benefit from the present study, learning which purposes of gamification can serve them. For example, gamification can (i) provide support in increasing health and wellness awareness among several classes of people (patients, wellness-interested people, and prevention-interested people) [103–105], (ii) ensure more effective training activities enhancing learning about matters or procedures (pharmacy, medicine, telehealth, therapies, and intervention protocols) [106–108], and (iii) increase the motivation of students, internships, and practitioners, especially in case of burnout [109,110].

## 6. Conclusions

This study explored the field of gamification and its specific application in the agri-food industry. Our analysis revealed that gamification was intended as a strategy/approach, tool, process, form of activities or services, trend, use of game elements or social mechanism. This resulted in it being applied in education, learning, health, wellbeing and behavioural change in food choice contexts. Gamification is applied in the agri-food industry for different purposes, all of which could have an impact on social sustainability, including stimulating healthy nutrition, improving education, encouraging technology use, brand engagement and consumer profiling, and facilitating learning, training and monitoring.

Moreover, we discovered eight core dimensions that underlie the concept of gamification in the agri-food industry: its health-educational role, co-creation processes, functional gaming purpose, monitoring mechanism, reward mechanism, stakeholders' supply chain engagement, real-life contextual procedures, and socially motivational change. The application of a rigorous and well-designed review methodology according to PRISMA guidelines (see Supplementary Materials) and the 6W model ensures the replicability of the study and increases its validity. The application of MEH and PCA is a way to identify and quantitively confirm the insights derived from content analysis, representing an advancement in the field. However, several limits can be discussed; the qualitative methodology used to identify the study sample refers only to the Scopus database. Systematic research would need a conjunction of contributions from different sources. However, given the exploratory focus of this study, only Scopus was used, with the authors fully aware that this represents a limitation of the research given the lack of availability of non-open access papers and the search query, which, deliberately left as generic as possible, might not include results directly related to the agri-food industry. Future studies might use more structured and in-depth queries and confirm or disconfirm these results, as well as the validity and reliability issues related to the identified word clusters.

As a follow up, our results could be confirmed through further investigation leveraging different methods such as quantitative and bibliometric analyses. The analysis could be focused on practitioners' or managers' definitions, complementing our analysis of the academic definitions. Moreover, it might also be interesting to check whether or not the transversal dimensions identified through the mixed procedure of this study differ from context to context, to identify more transcendental elements that are invariant with respect to the field of application concerned. The potential applicability of the proposed methodology to a qualitative dataset could be useful for evaluating the effectiveness of gamified systems or approaches in an operative agri-food business environment. In fact, case studies in which gamified systems or approaches were implemented provide the possibility to collect qualitative dataset (such as reports or surveys with open questions or transcribed interviews) particularly at the end of the implementation process when the gamification application is evaluated by managers or employees. PCA could lead to the discovery of the latent meaning of the collected text and, if accompanied by sentiment analysis, could highlight the sentiments contained in managers' or employees' words. Finally, as our results attest to the potential of gamification to impact society (stimulating or modifying community behaviours, affecting educational practices and learning, supporting customers' and stakeholders' engagement, encouraging technology usage, supporting branding and consumer profiling), it could be interesting to reflect on the possibility to make conduct more incisive gamification for social sustainability goals. This possibility could be realized by integrating gamification with other strategies, such as policy changes, education, and awareness-raising initiatives, that are able to enrich the consciousness of business and policy makers. Moreover, the knowledge of gamification-based systems and approaches and gamification's perceived benefits should be shared and made more accessible to all AFSC stakeholders, who include the small-scale farmers and low-income classes of consumers who probably cannot access the gamified technological systems and approaches. In conclusion, in order to ensure the achievement of inclusive and effective social sustainability goals, the agri-food industry should carefully plan and implement gamification.

**Supplementary Materials:** The following supporting information can be downloaded at: https://www.mdpi.com/article/10.3390/su15129503/s1, PRISMA 2020 Checklist.

**Author Contributions:** Conceptualisation, M.E.L.; methodology, M.E.L., M.M. and F.S.; software, M.M., F.S. and M.C.D.L.; validation, M.E.L., M.M., F.S. and M.C.D.L.; formal analysis, M.E.L., M.M., F.S. and M.C.D.L.; investigation, M.E.L., M.M., F.S. and M.C.D.L.; resources, M.E.L.; data curation, M.M., F.S. and M.C.D.L.; writing—original draft preparation, M.E.L., M.M., F.S. and M.C.D.L.; writing—review and editing, M.E.L., M.M., F.S. and M.C.D.L.; visualisation, M.M., F.S. and M.C.D.L.; supervision, M.E.L. All authors have read and agreed to the published version of the manuscript.

**Funding:** This research received no external funding.

**Informed Consent Statement:** Informed consent was obtained from all subjects involved in the study.

**Data Availability Statement:** No new data were created or analysed in this study. Data sharing is not applicable to this article.

**Acknowledgments:** This research was partially supported by: (I) "REFIN" intervention co-financed by the European Union through the Apulian regional plan (POR Puglia 2014–2020, Asse prioritario OT X "Investire nell'istruzione, nella formazione e nella formazione professionale per le competenze e l'apprendimento permanente"–Azione 10.4–DGR 1991/2018–Avviso2/FSE/2020 n. 57 del 13/05/2019–BURP n. 52 del 16/05/2019) and, (II) "CORVALLIS 4.0" intervention cofinanced by "Programma operativo FESR 2014–2020 Obiettivo Convergenza–Regolamento Regionale n. 17/2014–Titolo II capo 1–"Aiuti ai pro- grammi di investimento delle grandi imprese"–Contratti di Programma Regionali (Art. 17)" (Project code Y27GYF2–CdP 2014–20 CORVALLIS 4.0).

**Conflicts of Interest:** The authors declare no conflict of interest.

**Appendix A**

**Table A1.** Definitions of Gamification from literature review.

| Definitions | Source |
|---|---|
| Gamification is defined as "using game design elements in non-game contexts to motivate and increase user activity and retention". | [69] |
| Gamification is a novel way to engage users with the chatbot application. | [56] |
| The term "gamification" stands for the use of gaming elements and development mechanics in non-gaming environments. | [66] |
| Gamification is the use of game elements and design in non-game contexts to promote behavioural change, the tracking of individual behaviours, and the provision of visual feedback on progress in the game. | [82] |
| Gamification is a concept that makes use of social mechanisms such as social influence or interaction by applying game mechanics. | [111] |
| Gamification refers to the use of design elements, which are characteristic of games, in non-game contexts. | [70] |
| The approach of using individual game-design elements in non-gaming contexts, called gamification. | [68] |
| The concept of gamification involves improving the user's experience and engagement in non-game services and applications by using game factors. | [112] |
| The commonly accepted definition of gamification regards it as the application of game design elements in a non-game context and a service provided for consumers to engage in a gameful experience. | [90] |
| While gamification does not have one standard definition within the literature, in the context of behavioural change, it refers to the incorporation of game-based elements into an intervention setting, or a behaviour change tool, leading to an understanding of it as the incorporation of game elements and mechanics into intervention strategies. | [29] |
| Gamification is the main feature that may motivate the use of the mobile applications, in which the user gains points that can be converted into discounts in different shops. | [55] |
| Gamification is a design strategy that attempts to reproduce the interactive powers of games and imitate key gameplay features in non-game contexts without designing a complete game. | [83] |
| Gamification is the use of game elements in non-game contexts. | [113] |
| The gamification approach is the incorporation of game elements to promote competitiveness (rankings), goal-oriented actions (badge achievements) and social interaction aspects. | [54] |
| Gamification is the application of the elements of game designs in non-game contexts in order to increase interest, motivation, and enjoyment, and thus promote engagement. | [79] |
| Gamification is the application of game design principles for the purpose of engaging users with a chosen service or products. | [73] |
| The notion of gamification involves "The design approach of implementing elements (affordances, mechanics, technologies) familiar from games to contexts where they are not commonly encountered." | [84] |

**Table A1.** *Cont.*

| Definitions | Source |
|---|---|
| Gamification is defined as "a process of enhancing service with affordances for gameful experiences to support users' overall value creation". It is a form of service packaging where a core service is enhanced by a rule-based service system that provides feedback and interaction mechanisms to the user with an aim to facilitate and support the users' overall value creation. | [67] |
| Gamification is "the process of using game mechanics and game thinking in non-gaming contexts to engage users and to solve problems". | [59] |
| Gamification is suggested to be a strategy that can lead to a change for the better toward positive nutritional habits. | [100] |
| Gamification is the use of game design elements in non-game contexts. | [77] |
| Gamification is defined as the employment of game elements in a non-game context to improve the user's engagement. | [92] |
| Gamification can be defined as using characteristics of gaming (self-purposefulness and hedonistic use) with an ultimately utilitarian goal such as behaviour change or learning. | [91] |
| Gamification is the use of game elements and mechanics in non-game contexts. | [75] |
| Gamification can be considered a tool with which to create new kinds of healthy snack products that are developed in a co-creation process with the consumers and snack producers in order to increase willingness to taste and eat healthy food such as vegetables and berries among children. | [87] |
| Gamification refers to the design approach that implements familiar elements (affordances, mechanics, and technologies) games in contexts where these elements are encountered. | [78] |
| Gamification is an engagement technique that can motivate farmers to return empty containers of agrochemicals in the reverse supply chain. Gamification is a process used to account for the correct movement of agrochemical packages and their registration in the blockchain and it is applied in this work as a mechanism to encourage participation and interaction among supply chain participants. | [89] |
| One method for enhancing participant interest and engagement is gamification; that is, incorporating elements such as feedback (e.g., sounds or graphics), rewards for satisfactory performance and a unified storyline. | [88] |
| The concept of gamification concerns a set of activities and processes, which are used to solve problems related to learning, through game mechanics. | [85] |
| Gamification can be understood as using game-typical elements in a non-game-typical context, e.g., to motivate customers towards purchasing a product. | [76] |
| Gamification is a process of adding game features that systematically stimulate the usage of the target system in order to boost overall value creation. | [114] |
| Gamification is an approach for enhancing learning among the young generation due to the assurance that it provides students and enhances their learning processes and results. | [57] |
| Gamification is defined as the implementation of the most common and entertaining mechanics of videogames, in contexts not related to videogames, and stems from the belief in its potential to encourage motivation, behavioural changes, friendly competition and collaboration in different contexts, such as customer participation, employee performance and social loyalty. | [80] |
| Gamification means using game design elements in non-game contexts. | [74] |

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
