# Peer review of "The Potential of Gamification for Social Sustainability: Meaning and Purposes in Agri-Food Industry"

_sustainability, doi:10.3390/su15129503_

Round 1

Reviewer 1 Report

Dear authors, I hope my comments will be useful and will let you improve the article.

The review of the article “Achieving social sustainability goals in agri-food industry: the potential of gamification strategies”

The article analyses definitions of gamification and identifies the core dimensions that underlie the concept of gamification. The current article uses the Meaning Extraction Method procedure (MEM) for creating a dataset and Principal Component Analysis (PCA) performed for the extraction of factors. The authors stated that eight core dimensions were detected. However, the article has several major shortcomings.

1. The title of the article has no relation to the content of the article. The article has nothing about sustainability, especially social sustainability, and its goals or reaching these goals. Thus formally, the article does not fit the scope of the journal. Additionally, the article does not include gamification strategies – it analyses definitions of gamification.

2. The title and the aim differ significantly. Even the aim presented in the introduction and the abstract are not the same. Assuming that the correct aim is presented in the abstract, there is some misunderstanding with the context – agri-food. In some places, authors say about the agri-food industry, somewhere – sector, context, supply-chain, applications, -oriented education or agri-food without anything at all. However, the authors did not explain what will be this agri-food and what will be outside. As a consequence, articles selected for data analysis were related to eating behavior [108] or nutrition and physical activity [43], package [109], food-energy-water (FEW) nexus [29], children with diabetes mellitus [95], fast-food restaurants [110] or even gastronomic tourism [42].

 3. The theoretical part (“Gamification strategies in the agri-food industry”) explains the benefits of gamification. However, the content of this part does not fit either the title of it or the aim of the paper. Probably it would be worth studying previous typologies, dimensions, and literature, related to the development of dimensions (see Sofia Marlena Schöbel , Andreas Janson & Matthias Söllner (2020): Capturing the complexity of gamification elements: a holistic approach for analyzing existing and deriving novel gamification designs, European Journal of Information Systems, DOI: 10.1080/0960085X.2020.1796531)

 3. Methodological part has many problems as well. The authors start this part with a description of the systematic search. The authors used just one database, while typically several databases are used. Keywords defined imprecisely – some articles in gamification use just names of gamification elements like badges, leaderboards, virtual achievements, performance feedback, and rewards or simply include the word „game“. Additionally the keyword „food“ does not fit the agri-food industry. Moreover, exclusion criteria included just one case, but gamification could be related to information technologies, design, etc.

4. Authors stated that 34 definitions of gamification made a total sample for the analysis. However these definitions were not related to the agri-food industry or somehow special – they repeated or paraphrased the definition developed by Deterding et al., 2011. Therefore, how it is possible to direct the results of the analysis to just one industry?

5. Another serious problem is related to trust in coding.  Figure 2 “Draft for the binary matrix adopted for the subsequent analysis” shows that authors used nominal codes but the codes were not binary (values 0,1 and 2) can be observed in Figure 2). However PCA could be used just with binary data, otherwise multiple correspondence analysis should be applied.

6. Some information from PCA analysis, related to data suitability was not included in the paper. However, based on the provided information we can see that more than 8 dimensions were extracted, but authors decided to use just 8 for some reasons. Moreover, the explained cumulative variance was less than 34%, which is quite lower than the accepted thresholds.

Due to the mentioned, we can’t rely on the results of the study and its relation to the agri-food industry.

Proofreading would improve easiness of reading. Plus higher consistence among sentences would be preferable. 

Author Response

Dear Reviewer, thank your for your suggestion. You can find in attachment our answer explaining the improvements to the Manuscript.

Reviewer 2 Report

Comments

1.      It’s a nice effort to use of gamification strategies to achieve sustainability goals in the Agri industry. However, it's important to note that gamification alone may not be enough to achieve social sustainability goals in the Agri-food industry. It needs to be complemented with other strategies such as policy changes, education, and awareness-raising initiatives. Moreover, there is a need to ensure that gamified systems are accessible to all stakeholders, including small-scale farmers and low-income consumers who may not have access to technology. Therefore, the implementation of gamification in the Agri-food industry needs to be carefully planned and executed to ensure that it is inclusive and effective in achieving sustainability goals.

2.      The message and idea of the paper are appealing. Furthermore, some language issues should be sorted out

3.      Data about results and policy implications need more clarification in the abstract.

4.      Introduction: It is well-presented and coherently developed if some typos or grammatical mistakes are handled with just a thorough read by the authors.

5.      Material and methods:

6.      Overall, I believe the authors' approach seems reasonable, however I do not believe my knowledge of statistical models is sufficient to comment in-depth on the methodology. Therefore, I will offer the authors some suggestions regarding the presentation of the manuscript contents and will defer to the other reviewers and journal editors to assess the methodological approach in detail. I have noted this in comments to the editors as well, but want to make the authors aware of why this review lacks comments on the methodology.

7.      Discussions: the authors fail to discuss the main important findings of the study and the implications of these findings for the policymakers.

8.      In Section Conclusion – the discussion about the study’s implications is not satisfactory. More discussion on why and how the results from this study would have implications for other industries and how to achieve other SDGs by using this strategy. Specifically, what lessons/strategies other industries and countries can learn from this study?

Author Response

(The authors gave the same response as above.)

Reviewer 3 Report

This paper reviews a number of different studies and provides a definition for gamification in the agri-food industry using bibliometric methods.

I found the paper so highly academic as to be almost impractical. In spite of the paper being about deriving a definition of gamification, I had to google gamification to get a working definition. I am still not clear how gamification is going to achieve social sustainability goals in the agri-food industry. How are digital platforms going to promote agri-food sustainability?

This is particularly pertinent given that the special issue the authors submitted to is entitled: “Digital Transformation of the Agri-Food Industry and Supply Chains to Take On the Challenges of Sustainability”.

Specific comments:

Section 2.

Line 131-152 Some of this is repeated in the results section (see lines 236-250).

Again, its not clear from section 2 how gamification will achieve the things mentioned here.

Section 4

Lines 252-269. This is methodology and does not belong in the results section.

Line 268. “The final elaboration provided 71 words.” I only counted 42 in table 1.

Section 7

Line 480-482. Why were only open access papers used? Do the authors not have access to subscription journals?

Can be improved

Author Response

(The authors gave the same response as above.)

Reviewer 4 Report

After reading the article entitled "Achieving social sustainability goals in agri-food industry: potential of gamification strategies” I formulated several observations:
1) The article well achieves the assumed research aim: identifying the core dimensions underlying the concept of gamification for agri-food.
2) The research methodology used is really valuable. It can be a hint for other researchers, especially in the area of professional bibliographic research or constructing theoretical definitions.
3) The strength of the text is the clearly described procedure of research.
4) The strength of the text is the use of quantitative methods in bibliographic research.
5) The strength of the text is that quantitative methods are the starting point for qualitative analysis and building synthesis of research.

Suggestions:
1) In my opinion, Chapter 2: "Gamification strategies in the agri-food industry" should be deleted. The purpose of this article is to define these strategies in the agri-food industry and it is well presented in the two last chapters. The information presented in Chapter 2 introduces unnecessary confusion.
2) Line 250 states that the synthesis of this analysis is not provided in the Appendix. I'm sure it's an oversight. In the Appendix, there are the results of the research, which the authors synthesize in Chapter 4: "Results".
3) The presentation of "Agri-food gamification strategies’ text-based PCA" (Table 1) is questionable to me: it is just a huge table with mostly empty cells. I understand the intentions of the authors. However, maybe it is possible to present this aspect of the research procedure in a different way.
4) I have doubts about the description of the practical use of research results by agri-food companies, agri-food policy makers, management engineers, technology makers, marketers and providers interested in gamification applications in the agri-food industry (lines 435-462). I believe that these are statements that go beyond the results of research, so they should be much more careful and balanced. As a result of the conducted research, a synthetic definition of gamification strategies in the agri-food industry was obtained, which is a great value of this work. In this fragment, a bit of force tries to break it down again.
5) If it is not an editorial requirement, I suggest reformulating the title: The potential of gamification strategies in the agri-food industry based on the Scopus publication (or using Principal Components Analysis). In my opinion, such a title would better reflect the content presented in the text.
6) It seems that using the presented in the article methodology could also be able to indicate the direction of evaluation of gamification strategy applications in the area of the agri-food industry. Did the authors consider such a direction for future research?

Author Response

(The authors gave the same response as above.)

Reviewer 5 Report

I believe that this paper revolves around the authors' enjoyment of using particular software or statistical techniques rather than on the usefulness of the study or its conclusions. A database search is not typically sufficient these days for the basis of a study, although if the study is a meta-analysis of the research to date on a topic, a database search would be how the data would be retrieved. But the point of that type of study would be to clarify the state of the knowledge about a certain topic. In this case, the authors are attempting to clarify the definition of gamification for this particular field. I cannot say whether or not this is useful to other researchers in the field, but I suspect that there are other topics that are more compelling than this.

While the meaning of the text is largely understandable, there are areas that need editing more than others to clarify what is being said. For example: "Gamification serves agri-food sector by several ways, as attested by a fed literature on theme." (lines 138-139) is both editable and confusing. The first part of the sentence could be rewritten as, "Gamification serves the agri-food sector in several ways", but the second half is not intelligible. I am unsure what a "fed literature on theme" means, although I expect that what the authors are saying is that they performed a search of the literature for information about how the agri-food sector was served by gamification. This example is indicative of the types of editing that need to be done on this article: editing for style (word choice, spelling, grammar, etc.) and editing for meaning (clearing up unreadable text). There are also various instances of typographical errors (for example "Deterding" and "Deterging" (lines 337 and 340) throughout the paper.

Author Response

(The authors gave the same response as above.)

Round 2

Reviewer 1 Report

Dear authors, thank you for your efforts to improve the paper. However,  many serious problems still exist.

The theoretical part does not exist at all.

The suitability of data for factor analysis is marginal since KMO=0,55 (if it is possible to believe in the calculation). Worth to notice that factorial analysis is not reliable when KMO<0,5.

No additional information on factorial analysis like communalities, etc. (required according to APA style of reporting).

No scientific arguments for the selection of just one database.

The article is not related to "social sustainability".

Using the term "gamification strategy" is misleading, since the research is based just on definitions of gamification.

Table 1 presents just a part of results. Nevertheless, the presented information is highly questionable because "competition" and "competitiveness" belongs to different factors as well as "participants" and "participation", etc.  

Some sentences like "gamification strategies are strategical in the agri-food industry to reach social and marketing goals" (see Abstract lines 12-13) look meaningless. 

Reviewer 5 Report

Thank you for the responses